

# Convective environment in pre-monsoon and monsoon conditions over the Indian subcontinent: the impact of surface forcing

Lois Thomas[1], Neelam Malap[1], Wojciech W. Grabowski[2,3], Kundan Dani[1], and Thara V. Prabha[1]

[1]Indian Institute of Trropical Meteorology, Pune, India
[2]National Center for Atmospheric Research, Boulder, Colorado, USA
[3]Institute of Geophysics, Faculty of Physics, University of Warsaw, Warsaw, Poland

*Correspondence to:* Thara V. Prabha (thara@tropmet.res.in)

**Abstract.** Thermodynamic soundings for premonsoon and monsoon seasons from the Indian subcontinent are analyzed to
document differences between convective environments. Pre-monsoon environment features more variability for both near-
surface moisture and free-tropospheric temperature and moisture profiles. As a result, level of neutral buoyancy (LNB) and
pseudo-adiabatic Convective Available Potential Energy (CAPE) vary more for the pre-monsoon environment. Pre-monsoon
soundings also feature higher Lifting Condensation Levels (LCLs). LCL heights are shown to depend on the availability of
surface moisture, with low LCLs corresponding to high surface humidity arguably because of the availability of soil moisture. A
simple theoretical argument is developed and showed to mimic the observed relationship between LCL and surface moisture.
We argue that the key element is the partitioning of surface energy flux into its sensible and latent components, that is, the
surface Bowen ratio, and the way Bowen ratio affects surface buoyancy flux. We support our argument with observations of
changes in the Bowen ratio and LCL height around the monsoon onset, and with idealized simulations of cloud fields driven
by surface heat fluxes with different Bowen ratios.
## 1  Introduction
Convective environment over the Indian subcontinent changes significantly from hot and dry pre-monsoon conditions to cooler
and wetter monsoon. The change comes from the dramatic evolution of the large-scale circulation (e.g.,Yin, 1949; Lau and
Yang, 1996 and references therein) that brings significant oceanic moisture during the monsoon. The conveyer belt of mois-
ture is the monsoon low-level jet (Joseph and Sijikumar, 2004) that moistens land areas, changes cloud characteristics, and
brings monsoon rains that are key to the Indian economy. The change from pre-monsoon to monsoon conditions is rapid with
convective precipitation driven by the surface heating in the pre-monsoon period giving way to an increase in cloud cover and
surface rainfall during the monsoon season (e.g.,Ananthakrishnan and Soman, 1988). Significant rainfall occurs over the west
coast and the northeastern region, and it further extends westward in association with the northwestward movement of weather
systems formed over the Bay of Bengal (Gadgil et al., 1984).
The monsoon low-level jet weakens during the monsoon break periods, influencing moisture content over land and strongly
reducing the rainfall (Sandeep et al., 2014; Balaji et al., 2017). Intraseasonal oscillations of monsoon rainfall are well docu-
mented (e.g., Goswami and Mohan, 2001; Gadgil, 2003) with active and break periods featuring considerable spatiotemporal



variations (Rajeevan et al., 2010). Initial studies of the monsoon boundary layers focused on the contrast between active and
break monsoon periods (e.g.,Parasnis et al., 1985) with a contrasting moisture availability in the lower troposphere. The ac-
tive/break monsoon conditions are characterized by lower/higher boundary layer heights (e.g., Kusuma et al., 1991). Higher
cloud bases also occur during weak monsoon conditions when lower atmosphere is drier compared to the active monsoon.
Parasnis and Goyal (1990) reports enhanced convective instability in the boundary layer on weak monsoon days when com-
pared to the active monsoon. Convective Available Potential Energy (CAPE), a proxy for the strength of convection, feature
higher values over coastal regions because of the presence of higher moisture in the boundary layer (Alappattu and Kun-
hikrishnan, 2009). That study argues that the temporal variability of CAPE and convective inhibition (CIN) is predominantly
controlled by the boundary-layer moisture. Resmi et al. (2016) shows that sustaining convective storms in the diurnal cycle
is possible because of moisture advection and increase of CAPE over rain shadow region of the Indian subcontinent. Diurnal
variations of CAPE are directly linked to water vapor content near surface, with higher CAPE environments favoring higher
precipitation (Balaji et al., 2017). Precipitable water (PW) and lifting condensation level (LCL) derived from various obser-
vations are also closely related (Murugavel et al., 2016). Balaji et al. (2017) shows that high PW conditions correspond to
shallower boundary layer (with boundary layer height close to LCL) and higher LCL combined with deeper boundary layer
height typically occur during drier conditions. Balaji et al. (2017) also illustrates diurnal variations of PW and CAPE during
wet and dry regimes within the monsoon.
The monsoon onset marks a striking change in the surface and boundary layer conditions because of the change of the parti-
tioning of the surface energy flux into its sensible and latent components. However, there are no comprehensive comparisons of
pre-monsoon and monsoon thermodynamic environments and their contrasting characteristics with respect to parcel buoyancy
and boundary layer characteristics. The soil moisture variations typically follow rainfall patterns or variations. Transition from
pre-monsoon to monsoon conditions is associated with increase in soil moisture (Sathyanadh et al., 2016) and thus with the
change of the partitioning of the surface energy flux into its sensible and latent fluxes. The ratio of the sensible to latent surface
heat fluxes is commonly referred to as Bowen ratio. Bowen ratio affects surface buoyancy flux that drives boundary layer dy-
namics (e.g., Stevens, 2007 and references therein) and affects the rate at which convective boundary layer deepens. It also sets
the mean boundary layer humidity (e.g., Ek and Mahrt, 1994), impacts the efficiency of moist convection heat cycle (i.e., the
ratio between mechanical work and energy input at the surface; Shutts and Gray, 1999) and the distribution of shallow convec-
tion cloud base mass flux (Sakradzija and Hohenegger, 2017). One thus might expect different boundary layer characteristics in
surface-forced pre-monsoon and monsoon conditions due to different Bowen ratios for the two environments. However, Bowen
ratio does not seem to affect the updraft intensity in deep convection (Hansen and Back, 2015). Instead, the free-tropospheric
conditions, impacted by larger-scale atmospheric dynamics, may affect the strength of convection as measured by parameters
such as CAPE, LCL height, or maximum pseudo-adiabatic parcel buoyancy.
Present study contrasts pre-monsoon and monsoon environments by analyzing a large set of soundings released from Pune,
India, in the semi-arid Western Ghat mountains rain-shadow region. Traditional measures of convective environment are dis-
cussed with the emphasis of surface forcing. Since no surface flux information is available for the region where long period
soundings were obtained, we use data collected at another location in the rain shadow area to document changes in the Bowen

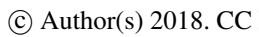



ratio and LCL height between pre-monsoon and monsoon conditions. Subsequently, we discuss two sets of idealized numerical
simulations that consider the impact of the surface Bowen ratio on convective development. We argue that model results are
broadly consistent with our interpretation of the sounding analysis. A brief summary concludes the paper.
**2  Observations**
**2.1  Data and Instrumentation**
Data from radiosonde measurements conducted at Pune (18°31′ N, 73°51′ E, elevation 530 m amsl) and measurements from
Mahabubnagar (16.75°N, 78.00°E, elevation 498 m amsl, about 500 km south-east of Pune) are used in this study. Both
locations are in the leeside of Western Ghat mountains in the semi-arid rain-shadow region. Total of 84 soundings from years
2010-2014 from Pune, divided into 42 pre-monsoon (March, April, May) and 42 monsoon (June, July, August, September)
soundings were analyzed. The Pune soundings are launched irregularly (typically once a week) and they are not part of the
daily global sounding network (i.e., they are not available, for instance, from the Wyoming air sounding database; http://
weather.uwyo.edu/upperair/sounding.html). The original data are archived at Indian Institute for Tropical Meteorology (IITM)
and they feature high spatial resolution as explained below. Vaisala radiosonde RS92-SGP is used, measuring atmospheric
temperature, pressure and humidity. Wind speed and direction (not considered in this study) are obtained by tracking the
position of radiosonde using GPS. Launch time is 13:00 IST when the solar insolation is near its peak. The operation takes
almost two hours with the radiosonde reaching typically up to 30 km altitude with an ascent rate around 5 ms$^{-1}$. Data is
available at approximately 3 m vertical resolution.
Second set of observations are surface flux and tropospheric profiles from the Integrated Ground Observational Campaign
(IGOC) at Mahabubnagar, south-east of Pune. Observations were conducted during the transition from pre-monsoon to mon-
soon and during the monsoon season of 2011. The latent and sensible surface heat fluxes were measured using eddy covariance
sensors located on a meteorological tower at 6 m above surface. In addition to the surface heat flux measurements, a microwave
radiometer profiler (MWRP) was also placed about 1.2 km from the tower location. MWRP provides vertical profiles of tem-
perature and humidity during the diurnal cycle (Balaji et al., 2017). This information is used to calculate lifting condensation
level applying the same method as for the Pune soundings (see the next section).
**2.2  Analysis of Pune soundings**
For Pune soundings, thermodynamic parameters such as the potential temperature ($\theta$), equivalent potential temperature ($\theta_e$),
water vapour mixing ratio ($q_v$), relative humidity ($RH$), cloud water mixing ratio ($q_c$), parcel buoyancy ($B$) and cumulative
Convective Available Potential Energy ($cCAPE$) were derived using thermodynamic equations and standard procedures as
described below. Standard parameters describing convective environment, such as lifting condensation level (LCL), level of
free convection (LFC), and level of neutral buoyancy (LNB) were calculated as well.





Pressure ($p$), temperature ($T$), water vapour mixing ratio ($q_v$) and relative humidity ($RH$) of the environment were given as
the standard sounding data. Geometrical heights of data levels were obtained by integrating the hydrostatic pressure equation
from the surface upwards. Subsequently, the input data were interpolated to a regular vertical grid with a uniform spacing of
50 m. A simple adiabatic parcel model was then applied to calculate various parameters describing convective environment.
Initial conditions for parcel came from lowest levels available in the sounding, typically corresponding to the near-surface
conditions. The potential temperature, water vapour mixing ratio, cloud water mixing ratio and the pseudo-adiabatic buoy-
ancy ($B$) inside the parcel (i.e., neglecting cloud water which is assumed to convert to precipitation and fall out) were de-
rived considering only condensation of water vapor. Condensation was calculated assuming that the parcel maintained water
saturation and corresponding latent heating was added to parcel potential temperature. The first level where condensation oc-
curred was marked as LCL. The level above LCL where parcel buoyancy became positive was marked as LFC, and the level
where parcel buoyancy changed from positive to negative (typically in the upper troposphere) was marked as LNB. Pseudo-
adiabatic parcel buoyancy was calculated as $B = g(\Delta\theta_v/\theta_{ve})$ where $\theta_v$ and $\theta_{ve}$ are virtual potential temperatures of rising
parcel and of the environment, respectively, and $g$ is gravitational acceleration. The virtual potential temperature is defined as
$\theta_v = \theta(1 + \varepsilon q_v)$, where $\varepsilon = R_v/R_d - 1 \approx 0.61$ and $R_v$ and $R_d$ are gas constants for the water vapour and dry air, respectively.
The cumulative CAPE ($cCAPE$) was calculated by vertical integration of the parcel positive buoyancy; it is formally defined
as $cCAPE(z) = \int_0^z max(0, B)dz$. Cumulative CAPE shows how CAPE builds up within a rising pseudo-adiabatic parcel.
Note that CAPE is given as $cCAPE(z = LNB)$. In addition, the equivalent potential temperature $\theta_e$ was calculated using
approximate formula:
$$\theta_e = \theta\, exp\left(\frac{L}{C_p T}\, q_v\right) \qquad\qquad (1)$$
where $L$ is latent heat of condensation.

## 110   3   Results of sounding analysis

### 111   3.1   Temperature and moisture profiles

Figure 1 shows vertical profiles of the potential temperature, water vapor mixing ratio and corresponding relative humidity
separated into pre-monsoon and monsoon periods. Panels with the potential temperature profiles also show corresponding
LCL levels. The atmosphere exhibits contrasting features during the two seasons that are discussed below.

### 115   3.1.1   Temperature and moisture profiles and their variability

For pre-monsoon conditions, the surface temperature is on average several degrees warmer and water vapor mixing ratio is on
average about half of that for monsoon period. The latter is arguably related to the contrasting levels of soil moisture in pre-
monsoon and monsoon conditions. The temperature and moisture profiles exhibit less day-to-day variability for the monsoon
period. The spread of temperature in middle troposphere in the monsoon environment is about half of that for pre-monsoon.
In upper troposphere and lower stratosphere, the differences are smaller. For the pre-monsoon period, moisture profiles below





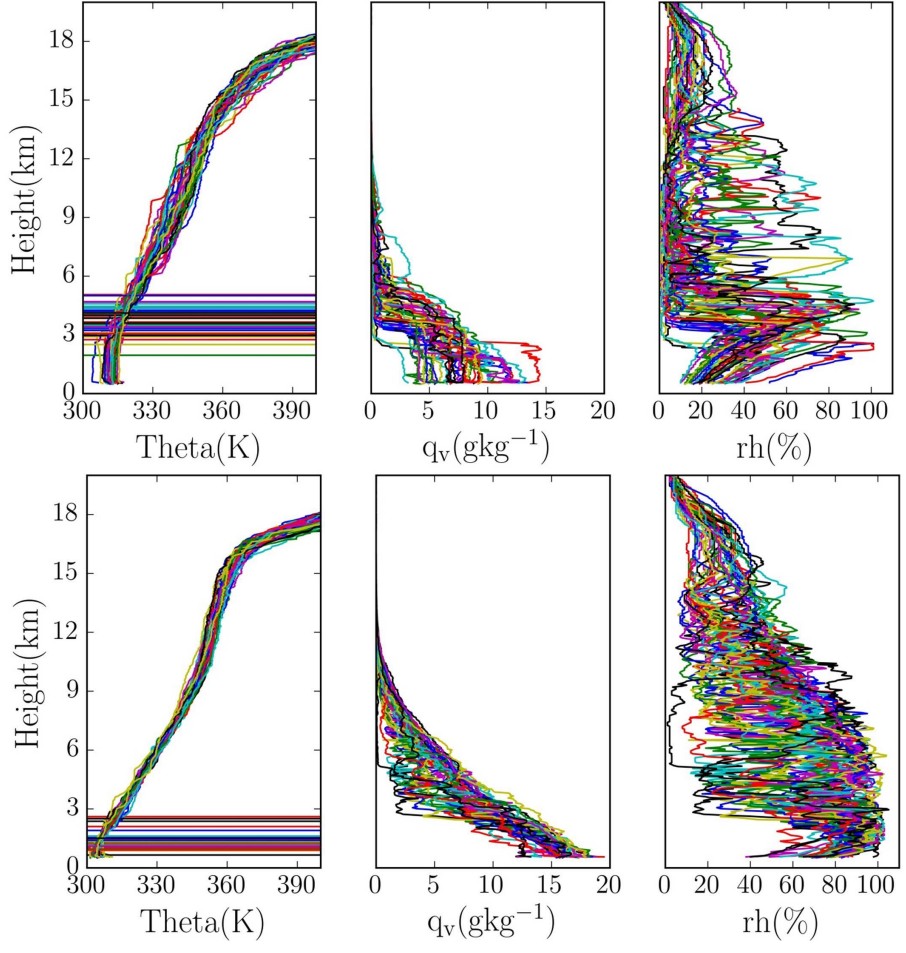

**Fig. 1.** Profiles of potential temperature (left panels), water vapor mixing ratio (middle panels), and relative humidity (right panels) for pre-monsoon (upper row) and monsoon (lower row) soundings.

6 km vary significantly and atmosphere is significantly drier above 6 km when compared to monsoon soundings. Arguably,
higher moisture contents in the middle and upper troposphere during monsoon come from convection reaching higher levels
as documented later in the paper. However, differences due to large-scale horizontal advection may play some role as well.
Individual moisture profiles feature significant fluctuations, even more apparent if no smoothing is applied to the original
high resolution data. This is evident at lower levels (i.e., within the boundary layer) as well as aloft. Fluctuations within
boundary layer show that it is not well-mixed for the water vapor in most soundings, especially for monsoon conditions.
However, relative humidity does increase approximately linearly within the boundary layer in most profiles similar to the case
of well-mixed mixed boundary layer (i.e., featuring constant with height potential temperature and water vapor mixing ratio).



### 3.1.2 LCL/boundary layer height

In surface-driven convective situations and mid-day conditions with either shallow or deep convective clouds above boundary layer, LCL height should be relatively close to boundary layer height as noted by Balaji et al., 2017 using temperature and moisture profile observations with microwave radiometer profiler. This is because the adiabatic (neutral) temperature profile (i.e., constant $\theta$) within the well-mixed boundary layer has to change to stably-stratified profile (i.e., $\theta$ increasing with height) in the free troposphere aloft. Since LCL marks the transition from dry to moist temperature lapse rate within a rising adiabatic parcel, the change from neutral boundary layer and moist-convecting stratified atmosphere aloft should also correspond to LCL. This is consistent with idealized simulations of the diurnal cycle of shallow and deep convection over land (see Brown et al., 2002 and Grabowski et al., 2006, respectively). These simulations show that deepening of boundary layer is accompanied by an increase of LCL height. However, presence of deep convection and significant precipitation can lead to the separation of the well-mixed boundary layer height and LCL height as illustrated later in the paper in idealized simulations (cf., Section 5). As Fig. 1 documents, LCLs around 13:00 LST are significantly higher for pre-monsoon period. This may come from either different surface fluxes during the course of the day between pre-monsoon and monsoon periods or from partitioning of the surface energy flux into sensible and latent components. One can argue, however, that the energy passed from earth surface to the atmosphere (the sum of sensible and latent heat fluxes) should be similar in pre-monsoon and monsoon conditions because the solar insolation is similar in both cases. Presence of extensive clouds in monsoon conditions can make a difference for surface energy budget, but we neglect this aspect for the qualitative discussion here. Thus, we assume that development of convective boundary layer during pre-monsoon and monsoon periods is to the leading order affected by partitioning of total surface energy flux into its sensible and latent components, and not by the differences in total flux.

The partitioning of surface flux into sensible and latent components depends on the soil moisture that differs drastically between pre-monsoon and monsoon conditions. The surface buoyancy flux that drives boundary layer dynamics is affected by the surface Bowen ratio. Since the thermodynamic variable relevant for buoyancy flux is the virtual potential temperature $\theta_v = \theta(1 + \varepsilon q_v)$, the total surface buoyancy flux BF can be approximated as $BF = <w\theta_v> = <w\theta> + \theta_o \varepsilon <wq_v>$, where $\theta_o$ is the surface potential temperature. Total surface energy flux EF can be similarly written (using the moist static energy or the equivalent potential temperature) as $EF = <w\theta> + \frac{L}{c_p} <wq_v>$. Consequently, BF/EF ratio between the buoyancy and energy surface fluxes can then be represented as:

$$BF/EF = (\alpha + B)/(1 + B) \qquad (2)$$

where $\alpha = \theta_o \varepsilon \frac{c_p}{L} \approx 0.1$ is a numerical coefficient, and $B = \frac{c_p <w\theta>}{L <wq_v>}$ is the Bowen ratio. For small Bowen ratios (i.e., surface latent heat flux dominates as typically over the oceans) the BF/EF ratio approaches 0.1, that is, only 10 % of the total surface energy flux contributes to the buoyancy flux. For large Bowen ratios (i.e., surface sensible heat flux dominates as over arid and semi-arid areas) the BF/EF ratio approaches 1, that is, all of the total surface energy flux contributes to the buoyancy flux. For Bowen ratio of 1 (i.e., equal surface sensible and latent fluxes), only about half of the energy flux contributes to the buoyancy flux. The equation (2) is shown in Fig. 2. The impact of the surface Bowen ratio on the shallow convective cloud base mass flux has recently been highlighted by Sakradzija and Hohenegger (2017).




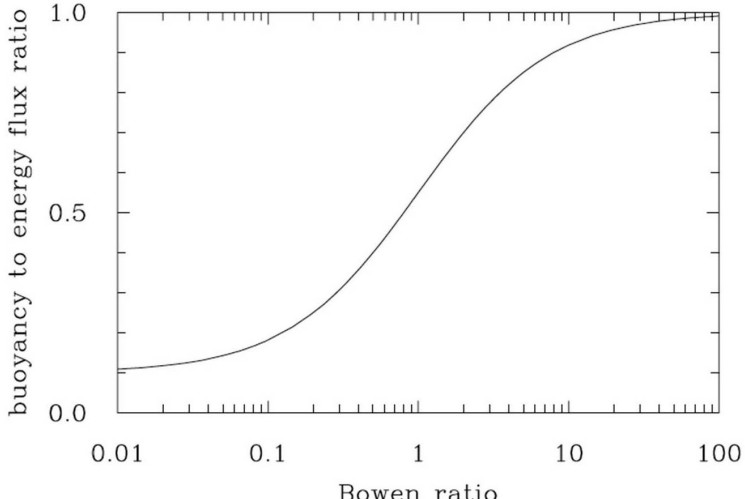

**Fig. 2.** The ratio of surface buoyancy flux to total energy flux (sensible plus latent) as a function of Bowen ratio.

The above considerations explain the well-known fact that daytime convective boundary layer develops deep over arid and
semi-arid areas that feature high Bowen ratio due to limited availability of water at the surface. For instance, over the Sahara
desert, the boundary layer height can reach several kilometres (e.g., Ao et al., 2012). In contrast, surface-driven convective
boundary layer over tropical and sub-tropical oceans is relatively shallow, often a mere several hundred meters. We argue
that the differences between pre-monsoon and monsoon periods can, to a large degree, be explained by the availability of soil
moisture and partitioning of surface energy flux between sensible and latent components. These differences will be further
illustrated by model simulations discussed in section 5.
**3.1.3   Troposphere-stratosphere transition**
As Fig. 1 illustrates, the tropopause is much better defined and varies less during monsoon. In contrast, transition from tropo-
sphere to the stratosphere is gradual in pre-monsoon environment. This may come from the fact that convection not always have
a chance to get to the tropopause in pre-monsoon environment (as documented later in the paper) and other processes (e.g.,
large-scale advection or radiative transfer) play an important or even dominant role. A well-defined tropopause is a feature
of the monsoon environment. This is associated with the mid tropospheric anticyclone of the Asian monsoon system. Dethof
et al. (1999) shows that the upper level monsoon anticyclone located close to tropopause is moistened by the monsoon convec-
tion. The strong potential vorticity gradients around tropopause prevent transport across upper-troposphere lower-stratosphere
(UTLS) region and result in a strong temperature gradients there. Pune latitude is in the region separating upper level westerlies
to the north and easterlies to the south that are associated with mid tropospheric anticyclone.
In the case of pre-monsoon conditions, the moisture availability in BL is considerably reduced and this has a significant
influence on cloud base height. Air parcels need to rise to greater heights in pre-monsoon conditions to reach LCL compared





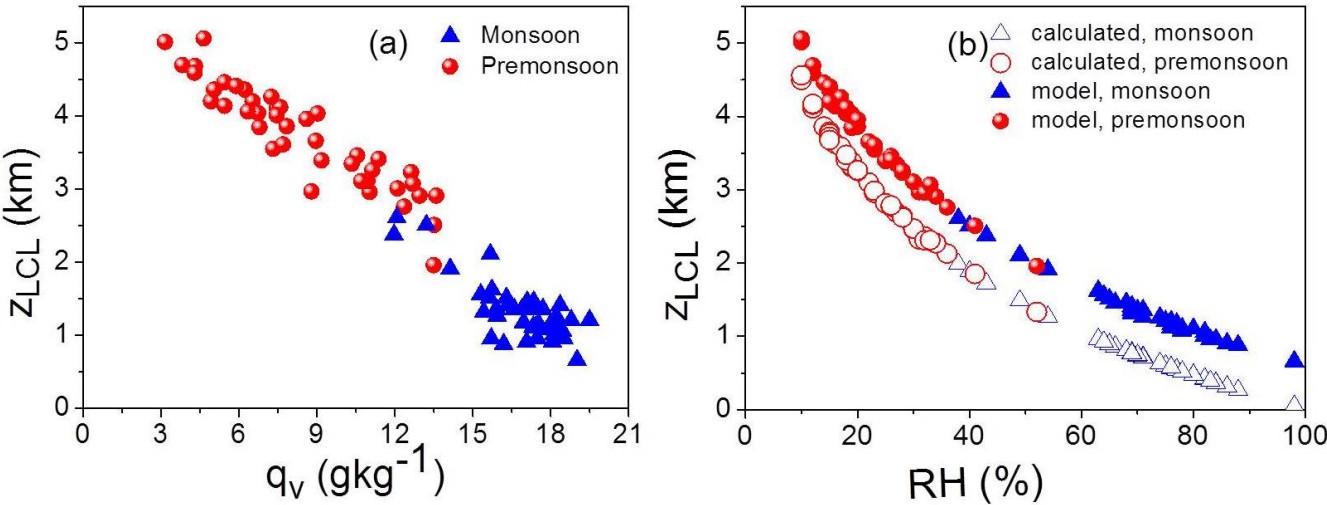

**Fig. 3.** Variation of LCL height $z_{LCL}$ with (a) surface $q_v$ and (b) surface $RH$. Red/blue circles/triangles represent pre-monsoon/monsoon cases. Parcel model derived parameters ($RH$ and LCL height) are shown as filled symbols. LCL heights derived using Eq. (3) are shown as empty symbols.

to monsoon conditions. Significant variations are observed in LCL heights during these two seasons. Pre-monsoon clouds have
their bases at higher levels, 2 to 6 km from the surface, whereas monsoon soundings indicate cloud bases at lower levels with
most of them being lower than 2 km. This result is highly correlated with surface level moisture as documented below.
BL as well as mid-tropospheric moisture for the two seasons exhibit contrasting characteristics. The mean tropospheric
moisture is higher for monsoon soundings. During monsoon, the surface values of $q_v$ are higher compared to pre-monsoon,
and most of them fall within the range of 14-18 gkg$^{-1}$. Pre-monsoon surface $q_v$ has a lower but wider range from 3 to 14
gkg$^{-1}$. Monsoon soundings also indicate higher levels of mid-tropospheric moisture. The main reason is south westerly winds
that transport moisture from Arabian Sea to Indian subcontinent. Because of Western Ghat mountains, the transport features
strong low-level convergence over Indian west coast. However, for the inland locations over rain shadow region, the jet core
level is seen at 1.5-2 km, just above the boundary layer. Arguably, boundary layer convection developing during the day pushes
jet layer to an elevated height.

### 3.2   Cloud base height and surface level moisture

For a well-mixed boundary layer, water vapor mixing ratio near surface is the main determining factor for cloud base height.
Figure 3 shows the scatterplot of the cloud base height and the surface moisture. Monsoon soundings with higher surface
mixing ratios correspond to lower cloud bases, and pre-monsoon soundings with lower mixing ratios have significantly higher
cloud bases. The relationship between mixing ratio at the surface and cloud base height is approximately linear but with a
significant scatter. However, the relationship between cloud base height and near surface $RH$ is nonlinear, with little scatter.





The following simple theoretical analysis explains the tight relationship between surface relative humidity and the cloud
base height as shown in Fig. 3. The key assumptions are that the boundary layer is well-mixed and the cloud base is not far
from boundary layer top. The two assumptions ensure that air parcels originating from near the surface and reaching LCL
insignificantly change their thermodynamic properties during their rise. Overall, these should be valid assumptions in surface-
driven convective situations. However, presence of significant precipitation can change this picture as documented later in
the paper. When the two assumptions are valid, then the height of the cloud base (i.e., LCL) is the level where adiabatic
air parcel rising from surface reaches saturation. If $T_{LCL}$ depicts the LCL temperature and $T_s$ and $RH_s$ depict temperature
and relative humidity at the surface, then $q_{vs}(T_{LCL})/q_{vs}(T_s) = RH_s$ Since $q_{vs} \approx e_s/p$ where $e_s$ and $p$ are the saturation
water vapor pressure and environmental air pressure, it follows that $e_s(T_{LCL})/e_s(T_s) = p_{LCL}/p_s RH_s$, where $p_{LCL}$ and $p_s$
represent pressure at the LCL and surface, respectively. Applying an approximate Clausius-Clapeyron formula in the form:
$e_s(T) = e_0 \, exp[\frac{L}{R_v}\,(\frac{1}{T_0} - \frac{1}{T})]$, where $e_0$ is the saturated water vapor pressure at the temperature $T_0$, leads to: $\frac{1}{T_s} - \frac{1}{T_{LCL}} =$
$\frac{R_v}{L} \, ln(\frac{p_{LCL}}{p_s \, RH_s})$. Using the dry-adiabatic relationship between $T_{LCL}$ and $T_s$ in the form $T_{LCL} = T_s - \frac{g \, z_{LCL}}{C_p}$ gives:
$$ln(\frac{p_{LCL}}{p_s RH_s}) = -\frac{L \, g \, z_{LCL}}{c_p \, R_v \, T_{LCL} \, T_s} \tag{3}$$
To show that the relationship is approximately valid for the data used in this study, we derived $z_{LCL}$ from observed $p_s$, $RH_s$
and $T_s$, and the parcel model derived $p_{LCL}$ and $T_{LCL}$. As the figure shows, Equation(3) provides $z_{LCL}$ estimates that are
lower than the $z_{LCL}$ calculated from the parcel model, and the difference between $z_{LCL}$ estimated from the parcel model and
derived from Equation(3) is typically around 600 meters regardless of the surface humidity.
There are at least two explanations for the underestimation of $z_{LCL}$ by Equation(3), both associated with the well-mixed
assumption for the boundary layer. The first one has to do with the presence of superadiabatic layer near the surface (i.e., the
potential temperature decreasing with height), clearly evident in many soundings shown in Fig. 1. With the surface temperature
higher than the mean boundary layer potential temperature, $z_{LCL}$ needs to be higher to keep $z_{LCL}/(T_{LCL}T_S)$ approximately
constant on the right-hand-side of (3) as $p_{LCL}/p_s$ can change little. Since 600 m corresponds to about 6 K along the dry adia-
batic lapse rate, such an explanation would imply that the air temperature change across the superadiabatic layer is universally
about 6 K in the sounding data used here. This does not seem inconsistent with at least some soundings shown in Fig. 1. The
boundary layer may also become not well-mixed (i.e., develop moisture and temperature stratification) because of precipitation
or low-level horizontal advection, presence of neither is possible to deduce from the available data. In convective situations,
significant surface precipitation is always accompanied by convective-scale downdrafts and boundary-layer cold pools. Since
the air in a cold pool typically comes from middle troposphere, the low-level water vapor mixing ratio inside the cold pool is
typically lower than on the outside (e.g., Tompkins, 2001). In such a situation, the boundary layer cannot be assumed well-
mixed and entrainment of boundary-layer air into a plume rising from surface would lead to plume dilution and thus to the
increase of LCL height. Moreover the parcel model constitutes a significant simplification of real atmosphere in which the
sonde is flown taking a Lagrangian path and cutting across different air columns.
The above analysis is consistent with results discussed in Murugavel et al. (2016). They showed that the column precipitable
water (PW), the vertical integral of water vapor density in the atmosphere, is a good predictor of LCL temperature and height



over the Indian subcontinent. Since the column PW is dominated by moisture in the lowest levels (and in the boundary layer
in particular), the mixing ratio near the surface should then be well correlated with LCL height as documented in Fig. 3.
The above results can also be used in reverse. The fact that, despite some offset, there is an almost a perfect relationship
between $RH$ and $z_{LCL}$ implies that mid-day boundary layer for all soundings considered in this study is of convective type,
that is, with close to the adiabatic potential temperature profile from above the superadiabatic surface layer up to the convective
boundary layer height and LCL.

### 239  3.3    Profiles of pseudo-adiabatic buoyancy and cCAPE

CAPE represents the energy available for moist convection and larger values of CAPE indicate larger potential for strong con-
vection. Figure 4 shows profiles of pseudo-adiabatic buoyancy (i.e., the difference in the virtual potential temperature between
pseudo-adiabatic parcel and the environment) and cCAPE from all soundings separated into pre-monsoon and monsoon con-
ditions. In addition, pre-monsoon soundings are divided into three groups (marked by red, blue, and green lines in left panels)
depending on CAPE values, with red/blue/green colors corresponding to low/medium/high CAPE values. This partitioning
will be used in the subsequent analysis. Monsoon and pre-monsoon environments exhibit distinct patterns. First, there is a
significant day-to-day variability for both environments as marked by the spread in profiles, but the variability seems larger
for pre-monsoon conditions. The variability is affected mostly by the surface water vapor mixing ratio as quantified later in
the paper. Large CAPE pre-monsoon soundings (green colour) are characterized by pseudo-adiabatic parcel maximum buoy-
ancies that are not different from their monsoon counterparts, but LNBs and CAPE values (evident from end points of cCAPE
profiles) are typically lower for the pre-monsoon environment.
Most of the monsoon pseudo-adiabatic buoyancy and cCAPE profiles follow a consistent pattern, as shown in right panels of
Fig. 4. These soundings maintain positive pseudo-adiabatic buoyancies up to the upper troposphere with CAPE values typically
between 1000 and 2000 $\mathrm{Jkg}^{-1}$, except for a few cases. This is different for pre-monsoon soundings that feature wide range
of maximum in-cloud buoyancies, with three distinct branches. The first branch represented by green lines follows a pattern
similar to monsoon cases, but with lower CAPE values and lower LNBs. The second branch, marked by blue lines, represents
intermediate soundings with CAPE typically between 500 and 1000 $\mathrm{Jkg}^{-1}$, and LNBs typically in middle troposphere. Red
lines represent the cases with low CAPE and LNB located in lower or middle troposphere.
These results show that monsoon season feature convective environments that are all similar and can be grouped into a
single family. In contrast, pre-monsoon season witnesses a wide range of atmospheric conditions and convection with diverse
properties, from situations with low-CAPE and LNBs in the lower and middle troposphere to situations with CAPE comparable
to monsoon environments and LNBs in the upper troposphere. One distinct feature of high-CAPE pre-monsoon category is that
the positive buoyancy increases steeply above LFC compared to the monsoon cases where buoyancy increased gradually above
the boundary layer. This is possibly due to the stark difference in moisture above LFC between pre-monsoon and monsoon
environments and its impact on the psuedo-adiabatic buoyancy.







**Fig. 4.** Profiles of the pseudo-adiabatic buoyancy (upper panels) and cCAPE (lower panels) for pre-monsoon (left column) and monsoon (right column) soundings. cCAPE profiles terminate at LNB. Pre-monsoon soundings are divided into three groups marked by red, light blue and green lines depending on the CAPE value.

## 3.4 CAPE, LNB and maximum buoyancy as a function of surface conditions

Figure 5 relates CAPE and LNB to the surface water vapor mixing ratio $q_v$. Using surface relative humidity instead of $q_v$ gives similar results (not shown). Despite significant scatter, the clear pattern is evident: low $q_v$ pre-monsoon environment is associated with the lowest LNB and CAPE, with $q_v$ as low as a quarter of the high-CAPE monsoon cases. Gradual increase of





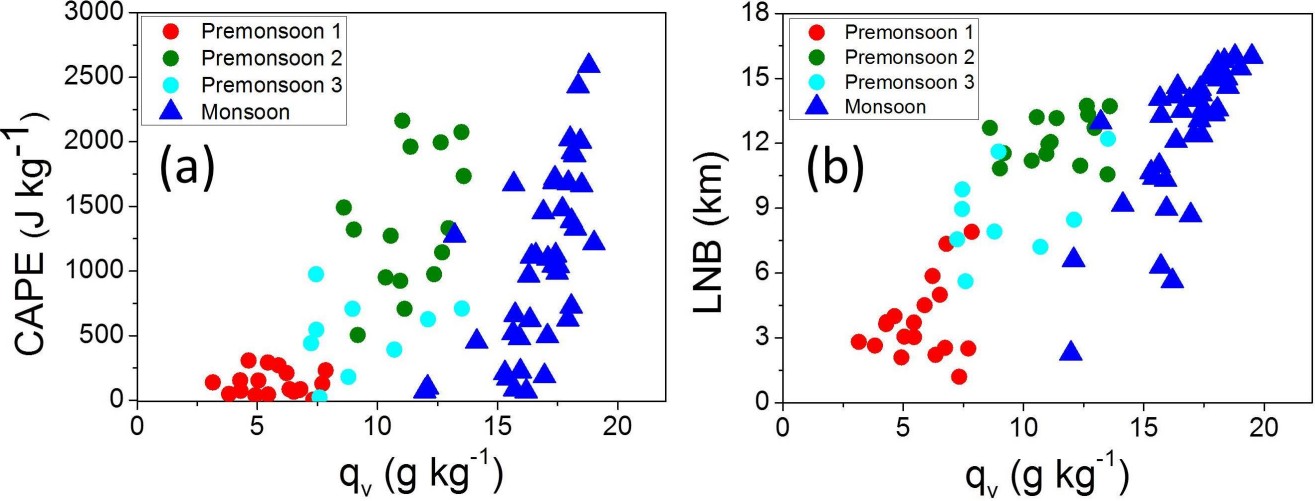

**Fig. 5.** Variation of (a) CAPE and (b) LNB as a function of surface water vapor mixing ratio. Pre-monsoon cases are grouped similarly as in Fig. 4

$q_v$ in pre-monsoon cases leads to gradual increase of CAPE and LNB. High CAPE and LNB monsoon cases are associated with
high surface $q_v$. The increase of CAPE with surface humidity is consistent with results reported in Alappattu and Kunhikrishnan
(2009), who analysed pre-monsoon observations over oceanic region surrounding the Indian subcontinent (cf., Fig. 8. therein).
Our study also supports findings of Bhat (2001), who reported that CAPE over Bay of Bengal during monsoon season varies
linearly with mixed layer specific humidity (cf., Fig. 3 therein). In our analysis, the linear relationship between surface $q_v$ and
CAPE is well defined for monsoon season, arguably because of the small free-troposphere temperature variations (cf., Fig. 1)
and small variations of LNB (Fig. 5b). Pre-monsoon convective environments exhibit larger scatter, arguably because of larger
variability of temperature profiles (Fig. 1) and LNBs (Fig. 5b).
Figure 6 shows the maximum pseudo-adiabatic parcel buoyancy as a function of surface water vapor mixing ratio, $q_v$ (panel
a) and the surface equivalent potential temperature, $\theta_e$ (panel b). Circles (triangles) mark pre-monsoon (monsoon) conditions
and the symbol colour depicts cloud base height according to the color scale shown to the right of the panels. Overall, neither
surface $q_v$ nor surface $\theta_e$ is a good predictor of the parcel maximum buoyancy. The maximum buoyancy does seem to increase
with the surface $q_v$, but the relationship is rather weak and there is a large scatter. The scatter reduced while soundings with
similar cloud base heights are considered.
The most apparent pattern, already discussed in section 3.2, is that the surface $q_v$ strongly affects the cloud base height. The
main contrast between pre-monsoon and monsoon conditions comes from contrasting relationship in low-level temperature
and humidity, that is, higher temperature and lower humidity for pre-monsoon cases, lower temperature and higher humidity
for monsoon cases. Because of compensating effects of the temperature and humidity on $\theta_e$, its surface values is thus not a
good predictor of the maximum pseudo-adiabatic parcel buoyancy either.



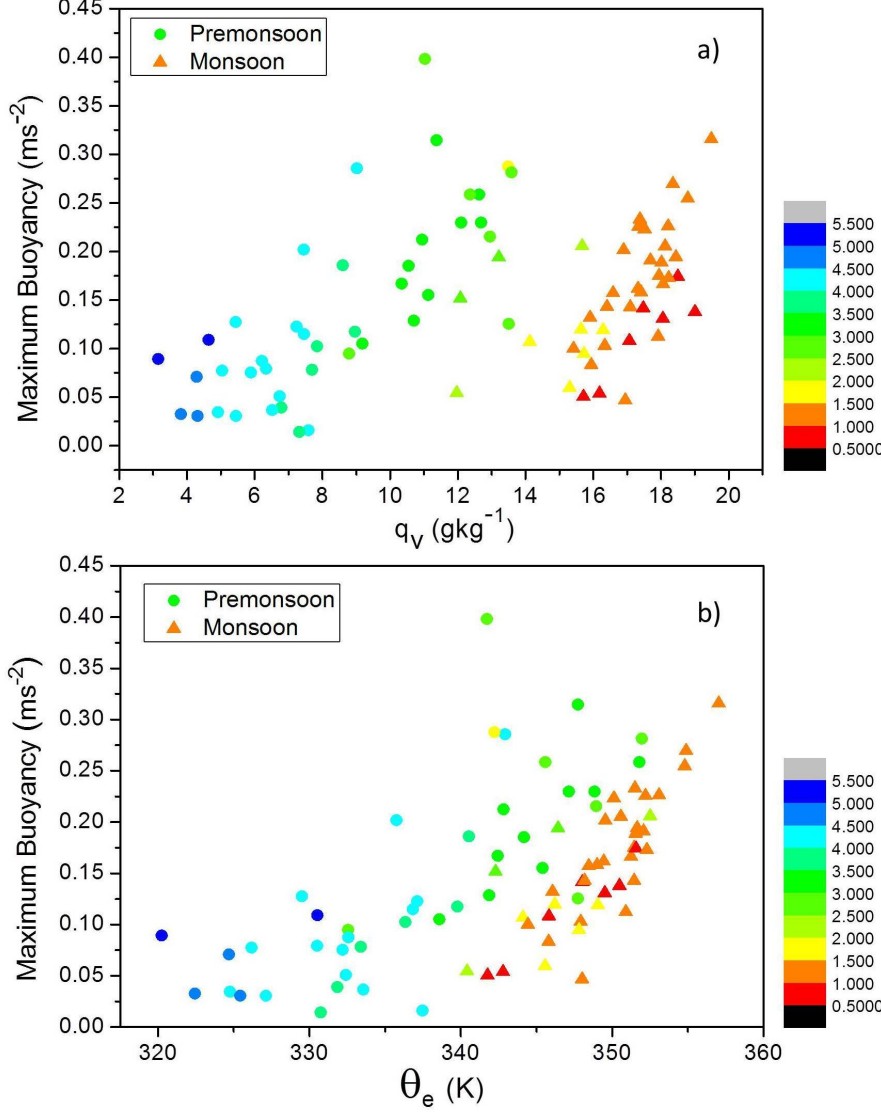

**Fig. 6.** Variation of the maximum quasi-adiabatic buoyancy as a function of (a) surface water mixing ratio and (b) surface equivalent potential temperature. Triangular/circular symbols are for monsoon/pre-monsoon soundings, with the color depicting the cloud base height. Pre-monsoon cases are grouped as in Fig. 4 and 5.

In summary, the availability of surface moisture seems to be a significant determinant of deep convection development
over Indian subcontinent (assuming that conditions near Pune can be taken as a representative for the rain shadow region),
with pre-monsoon and monsoon conditions providing contrasting examples of the impact. Day-to-day variability of surface
moisture is larger during pre-monsoon season and it adds to the variability associated with free-tropospheric conditions, such
as temperature and moisture stratification.

(c) Author(s) 2018. CC BY 4.0 License.



## 4 Observations of surface forcing during the pre-monsoon to monsoon transition

Since surface flux observations are not available simultaneously with Pune sounding data, we use observations collected during IGOC campaign to contrast the role of surface forcing between pre-monsoon and monsoon conditions. As explained in section 2.1, IGOC tower measurements of surface sensible and latent heat fluxes are combined with the estimates of LCL using MWRP-derived lower-tropospheric temperature and moisture profiles. Figure 7 shows evolutions of surface fluxes, Bowen ratio, and LCL height between June 24 and end of July using 3-hourly data during the diurnal cycle. Pre-monsoon to monsoon transition (monsoon onset hereafter) around July 1st is clearly evident in the figure. Before the monsoon onset, sensible heat flux is typically much larger than latent flux, and Bowen ratio is larger than 1. After monsoon onset, latent and sensible fluxes reverse, with latent heat flux becoming much larger than the sensible flux and Bowen ratio becomes smaller than one. The LCL height seems to decrease as Bowen ratio decreases after the monsoon onset and diurnal variations of LCL height become less significant after the monsoon onset. There seems to be a weak decreasing trend in the evolutions of Bowen ratio and LCL height after the monsoon onset, arguably consistent with the gradual increase of soil moisture during monsoon.

Although IGOC flux data shown here are for a single monsoon onset case, in contrast to 5 years of sounding data, the transition from the high-Bowen ratio pre-monsoon environment to the low-Bowen ratio monsoon environment is fairly typical over Indian subcontinent. The impact of surface Bowen ratio on the evolution of monsoon deep convection is further illustrated by numerical simulations discussed in the next section.

## 5 Simulations of deep convection driven by surface forcing

Two sets of idealized simulations of moist convection with emphasis on the surface forcing are discussed in this section in support of analysis presented previously. The first pair of simulations considers monsoon convection applying two specific mid-day soundings from IGOC field project, one corresponding to relatively moist surface conditions and second one for dry condition. The soundings are from the period toward the end of monsoon, 18th September (wet case) and 2nd October (dry case). As already explained, the soundings come from radiosonde released about 1.2 km away from surface flux tower site. The simulations are idealized because they apply mid-day sounding as initial condition and use mid-day observed surface conditions to calculate surface fluxes, driving the several-hour-long simulations. In reality, surface conditions change because of the diurnal variations of surface insolation.

Because of such a limitation, we employ a second set of simulations that considers a daytime convective development from an early morning sounding driven by evolving surface fluxes. The simulations are based on observations in the South American Amazon region (Grabowski et al., 2006). As an illustration, we introduce a simple modification of the surface Bowen ratio and analyze its impact. Although also idealized (i.e., prescribed horizontally-uniform surface fluxes), the simulations provide additional illustration of the role of surface forcing for deep convection development.



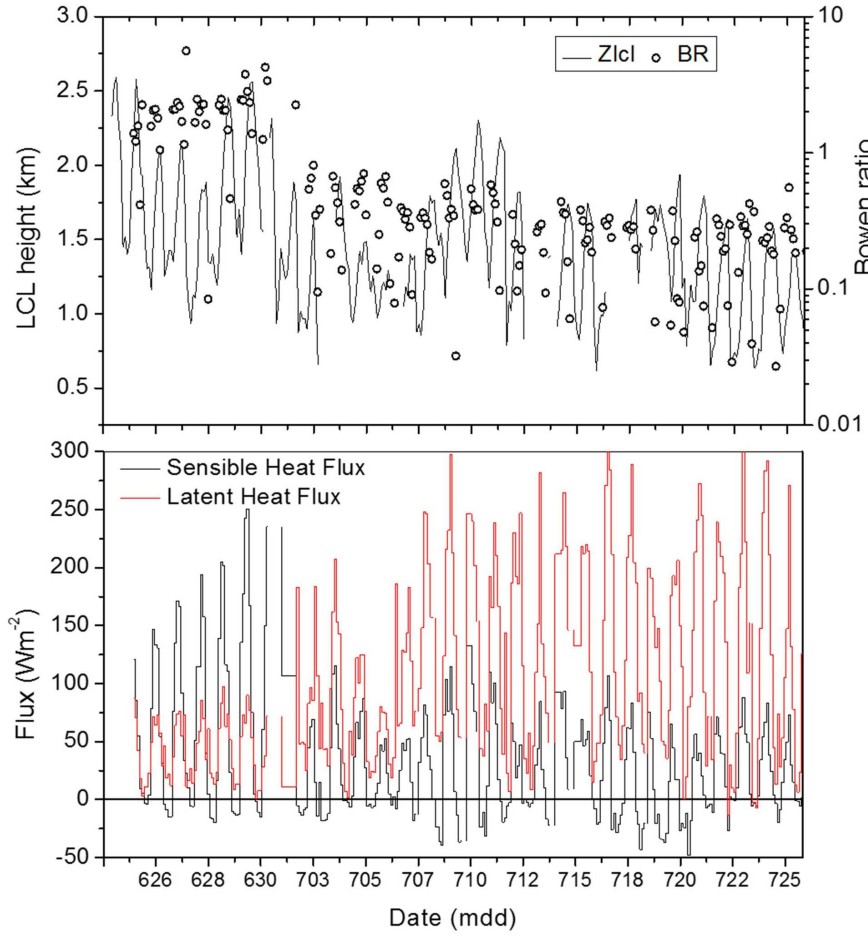

**Fig. 7.** Evolutions of (upper panel) Bowen ratio (BR) and LCL height ($Z_{lcl}$) and (lower panel) sensible (black lines) and latent (red lines) surface heat fluxes around the monsoon onset from the IGOC data.

## 5.1 Two IGOC cases of monsoon convection over India

Two contrasting soundings, referred to as wet and dry, were collected as the southwest monsoon was receding from Indian subcontinent and the lower atmosphere was getting progressively dry. The wet case is September 18th and the dry case is October 2nd. Soundings on both days were conducted around noon local time. The surface potential temperature and water vapor mixing ratio for the wet case were 305.2 K and 16.6 $\mathrm{gkg}^{-1}$. Corresponding values for the dry case were 306.1 K and 13.5 $\mathrm{gkg}^{-1}$. The contrasting surface temperature and moisture has been the determining factor for selecting these two cases.

Figure 8 compares the two soundings. Wet sounding features about 1 km deep mixed layer (although with a noticeable vertical moisture gradient) and relatively uniform free-tropospheric stability aloft. In contrast, dry sounding features no mixed layer near the surface, and a fairly complex structure in the lowest 5 km with distinct layers of approximately constant stability:





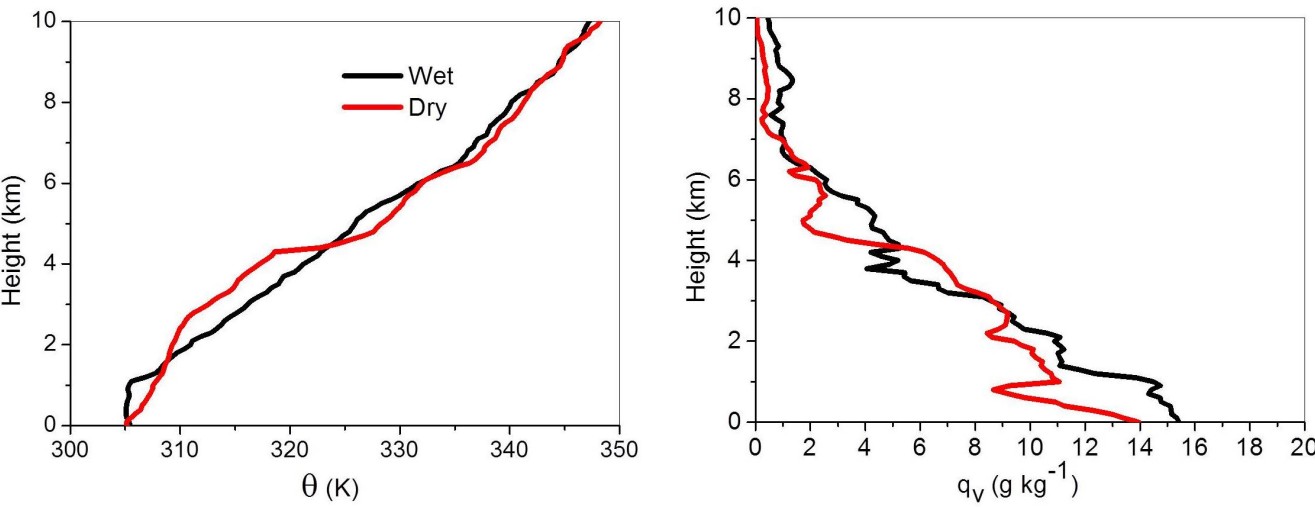

**Fig. 8.** Vertical profile of potential temperature and water vapour mixing ratio used in the simulations for wet and dry cases from the IGOC field project.

weakly stratified layer between the surface and about 3 km, typically-stratified layer between 3 and approximately 4.5 km, and an inversion between 4.5 and 5 km. The wet case has higher wind speed (greater than 4 $ms^{-1}$; not shown) compared to the dry case (smaller than 3 $ms^{-1}$). The mid-tropospheric inversion provides a barrier for deep convection as illustrated by model results. LCL height for the wet case is around 1.1 km, significantly lower than that for the dry case (around 1.6 km) due to moisture availability near the surface.

The model used for the two case simulations is the NCAR Weather Research and Forecasting (WRF) model (Skamarock et al., 2005) ran in the LES mode. The horizontal domain of 20 by 20 $km^2$ applies the 100 m grid length. The 10-km deep vertical domain is covered with a uniform grid with a 50-m vertical spacing. The model is run in an idealized manner for 8 hours applying surface fluxes derived from initial prescribed constant surface temperature and moisture values. Since the simulations start with horizontally-uniform conditions and require spin-up time to develop small-scale circulations and clouds, we present model results starting from hour 3. Figure 9 shows evolution of surface sensible and latent heat fluxes and Bowen ratio between hours 3 and 8. The sensible heat fluxes change little during the simulations, but latent fluxes decrease significantly, especially in the dry case. The initial total surface heat flux is about 20 $Wm^{-2}$ larger in the wet case and the difference increases as simulations progress. This implies that the surface total heat flux is larger in wet case and the difference between two simulations increases with time. The Bowen ratio is approximately 2 at the onset of two simulations. It remains close to 2 for the wet simulation, but increases to values around 12 at hour 8 for the dry case.

For the wet case, initial sounding features already a well-identifiable mixed layer (at least for the potential temperature), and the surface energy and the Bowen ratio change little throughout the simulation. Thus, the boundary-layer height increases steadily throughout the simulation, as shown in Fig. 10. The increase of boundary layer height in the wet case is accompanied by the increase of cloud base height. The depth of the cloud field, however, increases at a higher rate, from about 2.5 km at





**Fig. 9.** Evolution of (a) sensible heat flux, (b) latent heat flux, and (c) Bowen ratio in simulations of dry and wet cases from the IGOC field project. Red/black lines are for dry/wet case.

hour 3 to about 5 km at hour 8. For the dry case, mixed layer is absent in the initial sounding, and thus it rapidly develops
during the initial couple hours of the simulation. Boundary layer depth is about 1 km at hour 1 (not shown) and about 2.2 km
at hour 3. The rate then decreases significantly and boundary layer deepens subsequently at a rate comparable to the wet case,
about 100 m per hour. The cloud base height rises at a similar rate, and cloud field depth remains quite steady at around 2 km
between hours 3 and 8. The presence of a deep inversion between 4.5 and 5 km (see Fig. 9) provides an efficient lid for the
convective development.
The changes in cloud field between hour 5 and 8 are illustrated in Fig. 11 that shows corresponding cloud fraction profiles
for the two simulations. The figure illustrates increase of cloud base heights with time that are similar for dry and moist cases,





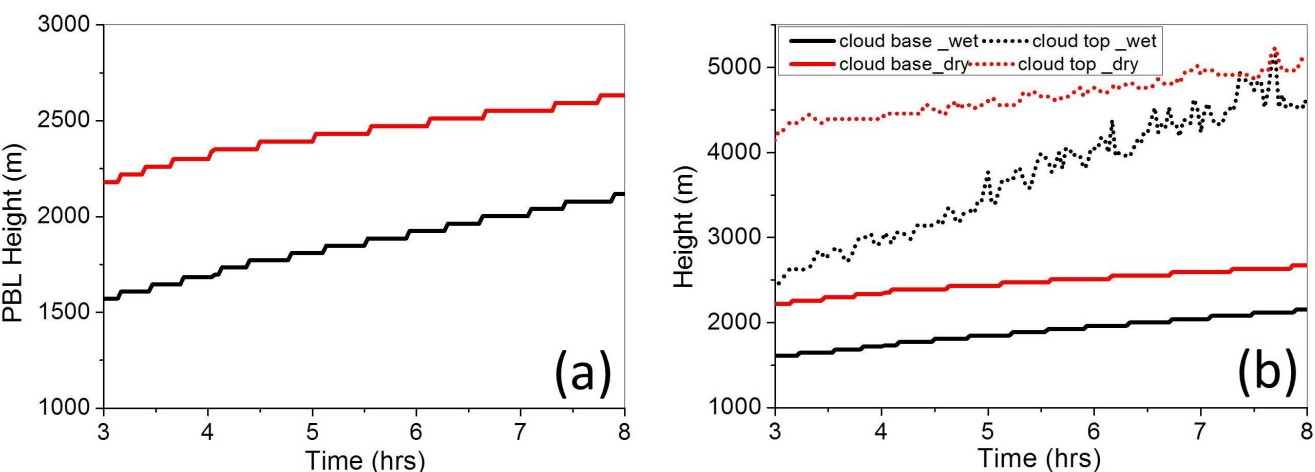

**Fig. 10.** Evolution of (a) PBL height and (b) cloud base and cloud top heights from dry and wet simulations of the IGOC field project. Red/black lines are for dry/wet case.

a significant deepening of the cloud field in moist case, and the impact of inversion between 4.5 and 5 km for the dry case that
results in almost 100 % cloud cover within the inversion.

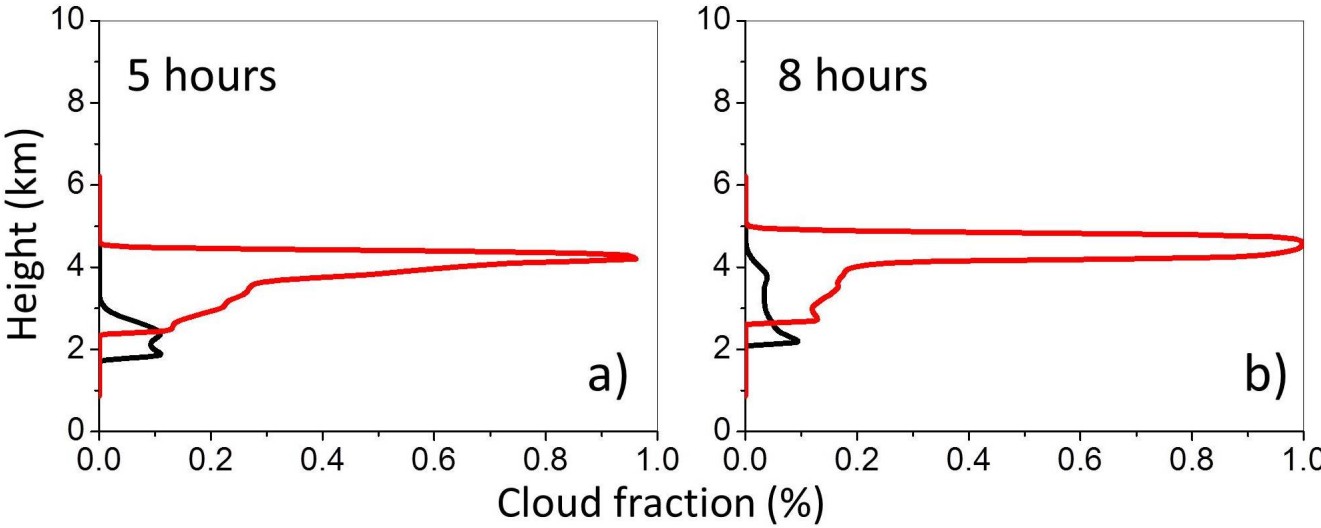

**Fig. 11.** Cloud fraction profiles at (a) hour 5 and (b) hour 8 from dry and wet simulations of the IGOC field project. Red/black lines are for dry/wet case.

In summary, high-resolution simulations of contrasting realistic cases observed over Indian subcontinent illustrate the impact
of surface forcing and highlight the role of specific free-tropospheric conditions for convective development and cloud fraction.



The latter are no doubt responsible to some spread of the observed convective environments apparent in Pune sounding data
analyzed in section 3.

## 5.2    Idealized simulations of daytime convective development over land: the LBA case

Since the first set of simulations applied highly idealized forcing, we use another pair of simulations that aim at simulating
daytime convective development over land, starting from the cloud-free morning sounding and finishing with the mid-day deep
convection. We apply the case developed in Grabowski et al. (2006) where observations from Amazon region motivated the
design of a simple modeling case. The case features formation and deepening of the cloud-free convective boundary layer in
the early morning hours, development of shallow convection in the late morning, and transition to deep convection around the
local noon. The 6-hour simulation covers period from 7.30 am local time (approximately at the sunrise) to the mid-day hours
(1.30 pm local time). It starts from horizontally-homogeneous morning sounding and is forced by increasing surface latent and
sensible heat fluxes mimicking effects of the increasing daytime surface insolation. This case has been used in several past
studies, such as Khairoutdinov and Randall (2006), Grabowski (2015) and Grabowski and Morrison (2016, 2017). We apply
the microphysical setup based on Grabowski (1999), that is, the one referred to as IAB in Grabowski (2015).
Two simulations are performed. The first simulation, referred to as LBA, follows original setup and features significantly
larger surface latent flux compared to the sensible flux, with the Bowen ratio between 0.4 and 0.5 as the surface fluxes evolve
(this is similar to wet cases during the Indian monsoon season). Surface fluxes are switched in the second simulation, that is,
the sensible flux takes values of the latent flux and the latent flux assumes values of the sensible flux. This simulation is referred
to as reversed LBA, or R-LBA, and it features the surface Bowen ratio between 2.0 and 2.5. According to Fig. 2, such a change
approximately doubles the buoyancy to energy flux ratio, from about 0.4 to about 0.7. One thus should expect significantly
deeper boundary layer to develop during the course of the R-LBA simulation.
The model used in the two simulations is the same as in Grabowski (2015) and Grabowski and Morrison (2016, 2017),
referred to as babyEULAG, a simplified version of the EULAG model (see http://www2.mmm.ucar.edu/eulag/). Since the
interest is in the boundary layer development, we apply a higher horizontal resolution with horizontal grid length of 200 m and
the same stretched vertical grid as in Grabowski (2015) and Grabowski and Morrison (2016, 2017). The horizontal domain
is 24 x 24 $km^2$. Overall, one can argue that differences between LBA and R-LBA towards the end of the simulation should
be relevant to the differences in the mid-day soundings between dry pre-monsoon and humid monsoon situations discussed
earlier.
Figure 12 and Fig. 13 summarize results of the two simulations pertinent to the impact of the surface flux Bowen ratio on
convective development. Figure 12 shows profiles of the cloud fraction in 1 hour intervals from 6-hour long LBA and R-LBA.
Overall, the profiles evolve in a quite similar way, with only shallow clouds at hour 2 and 3, and deep convection present at hour
5 and 6. The profiles at hour 4 correspond to the shallow-to-deep transition period. The differences in the cloud base height
in the simulations are apparent, with R-LBA (higher Bowen ratio) featuring higher mean cloud base. Figure 13 shows the
evolution of the mean cloud base height together with the evolution of the estimated height of the boundary layer. As the figure
shows, boundary layer depth is up to twice as deep in the R-LBA case than in the LBA case, especially between hours 2 and





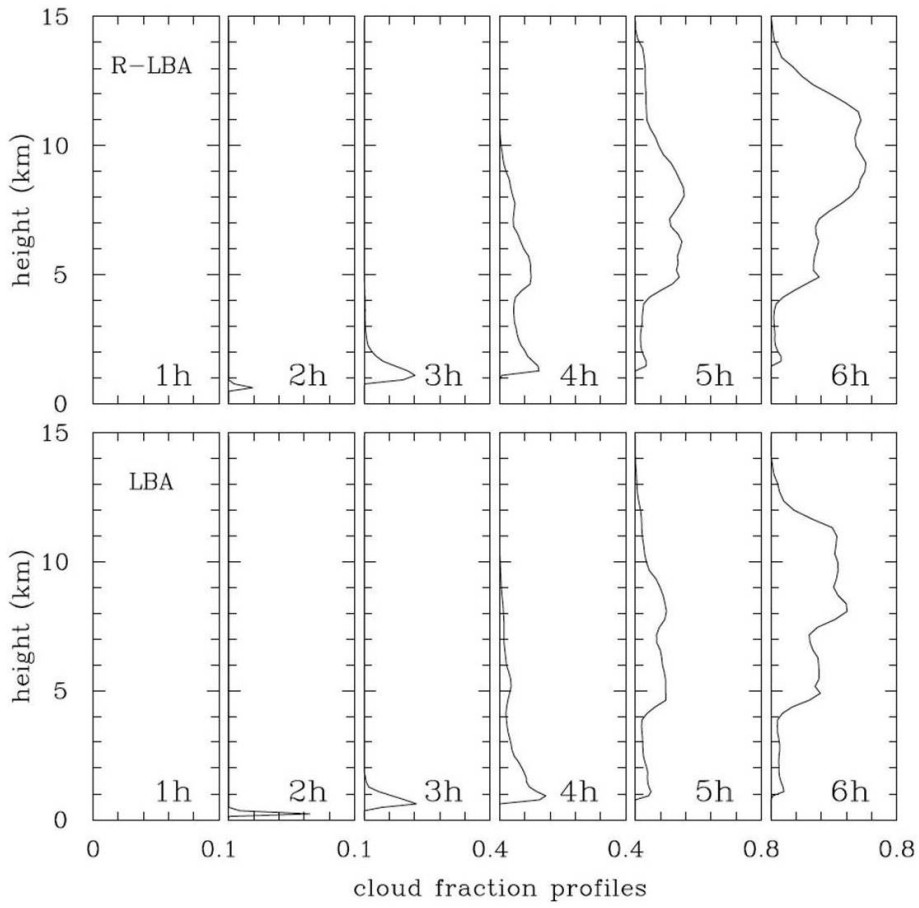

**Fig. 12.** Cloud fraction profiles at hour 1, 2, 3, 4, 5, and 6 from (upper panels) R-LBA and (lower panels) LBA simulations.

3 and during the hour 5 of these simulations. The cloud base height and the height of the boundary layer top track each other
well up to the onset of a significant precipitation after hour 3. The difference between the two heights is especially evident in
the LBA case as boundary layer height changes little during the two final hours. Specific differences between LBA and R-LBA
in the last two hours of the simulations may not be statistically significant due to the small domain size.
Overall, differences simulated in LBA and R-LBA cases highlight the impact of surface flux Bowen ratio and provide additional
support for its role in the difference between pre-monsoon and monsoon soundings.
**6   Summary**
Thermodynamic soundings released around local noon for several pre-monsoon and monsoon seasons over Indian subcontinent
were analysed. Various parameters, such as pseudo-adiabatic parcel buoyancy, Lifting Condensation Level (LCL), Level of
Free Convection (LFC), Level of Neutral Buoyancy (LNB), Convective Available Potential Energy (CAPE), cumulative CAPE



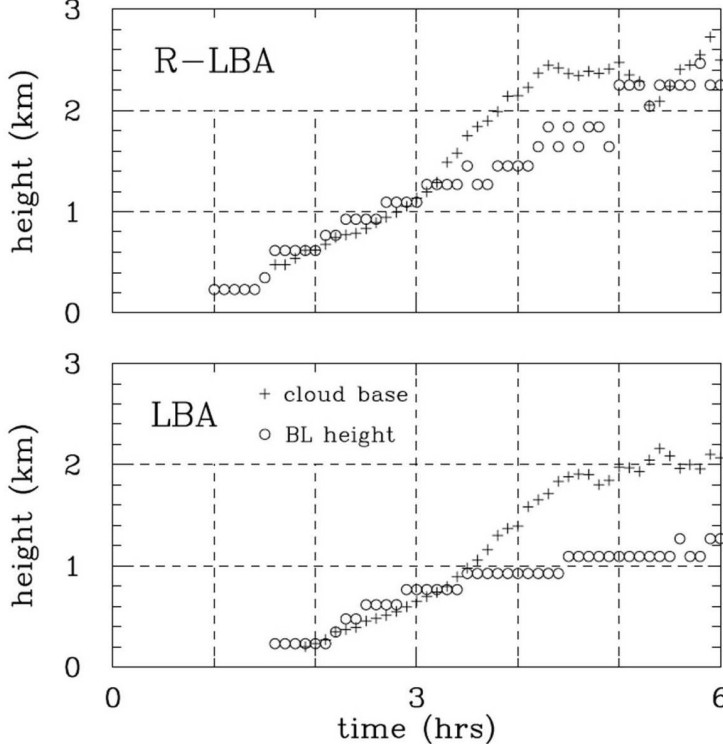

**Fig. 13.** Evolution of the cloud base height (plus signs) and the boundary layer height (circles) in R-LBA (upper panel) and LBA (lower panel) simulations. Dashed line are included to highlight differences between the simulations.

(cCAPE) were derived applying pseudo-adiabatic parcel model. Overall, pre-monsoon soundings show more variability of
surface and free-tropospheric conditions as documented in Fig. 1. For surface, the key is availability of surface moisture in both
pre-monsoon and monsoon environments, whereas variability of free tropospheric temperature and humidity for pre-monsoon
is arguably because of the impact of factors other than deep convection itself, for instance, the large-scale dynamics.

412       The pre-monsoon soundings feature higher cloud bases than monsoon soundings. We argue that this is a consequence of

partitioning of surface energy flux into its sensible and latent components as expressed by the Bowen ratio. For large Bowen
ratios (sensible surface flux is much larger than latent flux), the ratio between the buoyancy to energy flux is close to one, that
is, all surface flux contributes to buoyancy flux that drives boundary layer dynamics. For small Bowen ratios (i.e., sensible
surface flux much smaller than latent flux), only about 10 % of the energy flux is used for surface buoyancy flux (see Fig. 2).
We argue that the partitioning of surface energy flux into its sensible and latent components determines variations in the LCL
height as illustrated by observed rapid changes in Bowen ratio and LCL height near monsoon onset and illustrated in idealized
numerical simulations. Observations of LCL height and Bowen ratio during the pre-monsoon to monsoon transition illustrate
rapid and concurrent changes, with the Bowen ratio and LCL height decreasing significantly as the monsoon sets in. The impact





of surface forcing on the evolution of boundary layer and moist convection is also illustrated through numerical simulations
that complement sounding analysis.
The sounding data show that LCL height is linearly related to surface level moisture content with some scatter around the
perfect linear relationship (Fig. 3a). The scatter is eliminated when surface level relative humidity ($RH$) is used as a measure
of surface layer moisture content (Fig. 3b). A theoretical basis for such a relationship is developed, see Eq. 3. The theoretical
relationship between LCL height and surface level $RH$ mimics the relationship obtained with the parcel model. However, a
significant offset is present between the theoretical LCL height and LCL predicted by the parcel model. The offset is argued
to most likely come from the presence of surface superadiabatic layer not considered in the theoretical argument. The general
consistency between theoretical and parcel-model derived relationships between LCL height and surface moisture (Fig. 3)
supports the conjecture that surface forcing determines LCL height. This should be expected in high-insolation pre-monsoon
and monsoon conditions when surface forcing due to diurnal cycle drives formation of well-mixed convective boundary layer
in the morning and development of deep convection at later hours.
Overall, LNB and CAPE vary more for the pre-monsoon soundings. Large CAPE pre-monsoon soundings are characterized
by maximum pseudo-adiabatic parcel buoyancies that are similar to monsoon soundings. With a few monsoon exceptions,
low LNB and thus low CAPE soundings are present only for pre-monsoon environment. For both pre-monsoon and monsoon
soundings LNB and CAPE are linearly related to surface $q_v$ with a larger scatter for the pre-monsoon environment. In general,
neither surface $q_v$ nor surface $\theta_e$ are good predictors of the parcel maximum pseudo-adiabatic buoyancy, although there is a
general increase of the maximum buoyancy and CAPE with the increase of either the surface $q_v$ or $\theta_e$. The increase is along
different paths for pre-monsoon and monsoon soundings, see Fig. 4 and 5. The scatter is small for monsoon cases, no doubt
because of smaller variability of free-tropospheric structure as documented in Fig. 1.
Results presented in this paper should help understanding effects of aerosols, dramatically different between highly-polluted
pre-monsoon environment and relatively clean environment during the monsoon, on moist convection over Indian subcontinent.
Understanding dynamical effects, for instance, partitioning of the surface heat flux into its sensible and latent components and
how the partitioning affects cloud base height and cloud buoyancy, is required for a confident selection of deep convection
cases suitable for cloud seeding, the target of the ongoing Indian precipitation enhancement program (Prabha et al., 2011;
Kulkarni et al., 2012; Prabha, 2014).
*Acknowledgements.* This work was done as part of SN Bose Scholarship program (Indo-US Student Exchange program by IUSSTF) at
National Center for Atmospheric Research (NCAR) and M. Tech project work at Indian Institute of Tropical Meteorology (IITM). First
author thanks Department of Atmospheric and Space Sciences, Savitribai Phule Pune University for nominating to SN Bose Scholarship
program, and acknowledges NCAR and IITM for providing facilities to conduct the study. CAIPEEX experiment was funded by Ministry
of Earth Sciences. Authors acknowledge field contributions from several colleagues of Indian Institute of Tropical Meteorology, Pune in
collection of data used in this study. WWG was partially supported by the Polish National Science Center (NCN) "POLONEZ 1" Grant
2015/19/P/ST10/02596. The POLONEZ 1 grant has received funding from the European Union's Horizon 2020 Research and Innovation




Program under the Marie Sklodowska-Curie Grant Agreement 665778. WWG acknowledges IITM financial support and hospitality during
his visits to IITM.



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
