# Peer review of "Convective environment in pre-monsoon and monsoon conditions over the Indian subcontinent: the impact of surface forcing"

_Atmospheric Chemistry and Physics, 2018_

## Referee Comment (RC1) · Anonymous Referee #1 · 2 Feb 2018

The work examines the convective environment in the Indian subcontinent during the pre-monsoon and monsoon seasons using radiosonde and surface flux observations. Such data are highly valuable for this region. The analysis is robust and overall, the paper is well written. However, it lacks clarity at few places that need attention. These are highlighted below.

Fig.1 : Are the horizontal lines in left panels LCL? It's not possible to understand the pre-monsoon vs monsoon difference from these plots. I rather suggest to show mean (and the intra-seasonal spread) profiles or some other means to help following the discussion. Are there any major differences in active and break phase of the monsoon?

Fig 5 - For low qv case (pre-monsoon 1), CAPE does not show any variation; it starts showing some variation when qv is between 7-14 (pre-monsoon 2 case) and then for the pre-monsoon 3 case, t shows a linear behavior. Overall, the pattern looks exponential. Does this mean that there is a threshold qv above which CAPE responds to further change in qv? This needs to be clarified.

How do the two datasets match (e.g. do LCL from the two datasets match). This is important as the conclusions are based on both data sets.

More details about the numerical experiments are required. From where the boundary conditions are taken? What is the time step?

How are these results useful to understand the aerosol impact? Bringing aerosols further complicate due to forcing and feedback. I suggest to remove the reference of aerosols (last paragraph).

---

## Referee Comment (RC2) · Anonymous Referee #2 · 12 Feb 2018

This manuscript investigates thermodynamic soundings for premonsoon and monsoon seasons from the Indian subcontinent are analyzed to document differences between convective environments. Pre-monsoon environment features more variability for both near- surface moisture and free-tropospheric temperature and moisture profiles. As a result, level of neutral buoyancy (LNB) and pseudo-adiabatic Convective Available Potential Energy (CAPE) vary more for the pre-monsoon environment. The authors argue that the key element is the partitioning of surface energy flux into its sensible and latent components, that is, the surface Bowen ratio, and the way Bowen ratio affects surface buoyancy flux.

[Figure]

Overall, the manuscript is well written. It is obviously beneficial to have detailed analyses of observation data on the Indian monsoon. Idealized simulations are well setup. This reviewer, however, feel that the findings from the analyses are plain instead of new insights on the atmospheric physics related to Indian monsoon. For instance, it is very obvious to see that LCL heights are shown to depend on the availability of surface moisture, with low LCLs corresponding to high surface humidity arguably because of the availability of soil moisture.

1) The argument with observations of changes in the Bowen ratio and LCL height around the monsoon onset is clear. But, in other sense, the Bowen ratio is a resulting parameter instead of a controlling variable. The authors need to be careful in describing the analyses.

2) Regarding the soil moisture feedback, there are numerous literature that describes the soil-moisture-precipitation feedback processes (e.g., Asharaf et al. 2012, Soil Moisture–Precipitation Feedback Processes in the Indian Summer Monsoon Season). It is recommended to cite these papers in explaining the physical mechanism, and an addition of a new insight from the previous literature.

---

## Author Comment (AC1) · 20 Mar 2018

*Responses to comments from Reviewer 1 (comments in bold italics, responses in regular font):*

*Rev : The work examines the convective environment in the Indian subcontinent during the pre-monsoon and monsoon seasons using radiosonde and surface flux observations. Such data are highly valuable for this region. The analysis is robust and overall, the paper is well written. However, it lacks clarity at few places that need attention. These are highlighted below.*

We thank reviewer for the valuable comments.

*Rev : Fig.1 : Are the horizontal lines in left panels LCL?*

Horizontal lines in the left panels represent LCL heights for each of the soundings. We apologize for not explaining that in the figure caption. The line colour represents LCL and it is same as that for the corresponding profile. The modified figure is shown below.

[Figure]

Figure 1: Profiles of potential temperature (θ), water vapour mixing ratio ($q_v$) and relative humidity (RH) for (upper panels) premonsoon and (lower panels) monsoon soundings. Different colours represent different soundings with a total of 42 soundings for both cases.  Horizontal lines in left  panels are LCL heights with the same colour as the corresponding profile.

*Rev : It's not possible to understand the pre-monsoon vs monsoon difference from these plots. I rather suggest to show mean (and the intra-seasonal spread) profiles or some other means to help following the discussion.*

We intend to highlight the following inferences about the premonsoon and monsoon seasons through Figure 1.
- Deeper boundary layer (BL) in premonsoon than monsoon can be identified by the region of constant potential temperature in the lower atmosphere up to 3 km from surface;
- Higher cloud base heights are present for premonsoon clouds and lower cloud base heights characterize monsoon clouds as shown by LCL heights;
- Presence of higher moisture in BL as well as mid-troposphere for monsoon conditions contrasts premonsoon environment that features drier BL and troposphere.
- Premonsoon BL is typically also topped by a strong inversion, which characterize a sudden decrease in RH within a few 100 meters
- Higher relative humidity for monsoon BL with values closer to saturation is evident.
- Well defined tropopause for the monsoon is identified from the sharp gradient of potential temperature at the height of $\cong 16$ km

Details of these are discussed in subsections 3.1.1 – 3.1.3.
We feel the dataset is not large enough to present mean and standard deviations for the two sub-sets. We feel the individual profiles are adequate to support the above inferences.

*Rev: Are there any major differences in active and break phase of the monsoon?*

Retrieving information regarding active and break periods of the monsoon is impossible using the data we have available for this study from a single location. This is because relevant studies concerning monsoon active and break periods (e.g., Pai et al, 2016; Rajeevan et al, 2010) introduces classification based on the weather properties at a larger region, namely the monsoon core region, which covers most of the central India. Our study considers high resolution radiosonde measurements where the data is collected over a single location (Pune, 18$^{o}$ 31' N, 73$^{o}$ 51' E) over Indian peninsula and with contrasting local surface forcing. We feel the data cannot be representative of the larger monsoon region. Monsoon is having a significant spatio-temporal variability and classifying such local data for active break conditions may not be appropriate.

*Rev : Fig 5 - For low qv case (pre-monsoon 1), CAPE does not show any variation; it starts showing some variation when qv is between 7-14 (pre-monsoon 2 case) and then for the pre-monsoon 3 case, it shows a linear behaviour. Overall, the pattern looks exponential. Does this mean that there is a threshold qv above which CAPE responds to further change in qv? This needs to be clarified.*

Yes, we agree. There appears to be a threshold as suggested by the Reviewer. This may possibly suggest the difference between shallow and deeper convection (e.g., congestus). We pointed this out in the revised manuscript.

*Rev: How do the two datasets match (e.g. do LCL from the two datasets match). This is important as the conclusions are based on both data sets.*

We are not sure what the reviewer is asking here. We hope the information below helps.
The radiosonde measurements for the parcel analysis are from Pune (18° 31' N, 73° 51' E) and data for model simulations are from Mehabubnagar (16° 45' N, 78° 00' E). Both locations are in the leeside of Western Ghat Mountains in the semi-arid rain-shadow region.

*Rev: More details about the numerical experiments are required. From where the boundary conditions are taken? What is the time step?*

We expanded the description of the simulations and hope the extended text satisfies the reviewer.

*Rev :How are these results useful to understand the aerosol impact? Bringing aerosols further complicate due to forcing and feedback. I suggest to remove the reference of aerosols (last paragraph).*

The literature concerning observations of aerosol effects on deep convection is full of claims that mix correlations with causality. The fact that pollution and deeper clouds occur together does not mean that pollution (e.g., as for the premonsoon) causes convection to be become deeper. In other words, one needs to separate effects of pollution from effects of meteorology. We believe that a study like ours that focuses on the meteorology is useful. That said, we do not want to dwell on this aspect and only bring it in the closing paragraph on the paper.

References mentioned above:

Rajeevan et al, 'Active and break spells of the Indian summer monsoon' , J. Earth Syst. Sci. 119, No. 3, June 2010, pp. 229–247

Pai et al, 'Active and Break Events of Indian Summer Monsoon during 1901-2014', Clim. Dyn., vol.46(11); 2016; 3921-3939

---

## Author Comment (AC2) · 20 Mar 2018

*Responses to comments from Reviewer 2 (comments in bold italics, responses in regular font):*

*Rev: This manuscript investigates thermodynamic soundings for premonsoon and monsoon seasons from the Indian subcontinent are analyzed to document differences between convective environments. Pre-monsoon environment features more variability for both near- surface moisture and free-tropospheric temperature and moisture profiles. As a result, level of neutral buoyancy (LNB) and pseudo-adiabatic Convective Available Potential Energy (CAPE) vary more for the pre-monsoon environment. The authors argue that the key element is the partitioning of surface energy flux into its sensible and latent components, that is, the surface Bowen ratio, and the way Bowen ratio affects surface buoyancy flux.*

*Overall, the manuscript is well written. It is obviously beneficial to have detailed analyses of observation data on the Indian monsoon. Idealized simulations are well setup.*

We thank the reviewer for the useful comments.

*Rev: This reviewer, however, feel that the findings from the analyses are plain instead of new insights on the atmospheric physics related to Indian monsoon. For instance, it is very obvious to see that LCL heights are shown to depend on the availability of surface moisture, with low LCLs corresponding to high surface humidity arguably because of the availability of soil moisture.*

We realize that our findings are not ground-breaking as they highlight relatively-well understood impacts of surface moisture on Bowen ratio and cloud base height. However, we are not aware of any studies of these impacts on the convection over the Indian subcontinent and the difference between premonsoon and monsoon convection, especially with high resolution radiosonde observations for a long period.

*Rev: 1) The argument with observations of changes in the Bowen ratio and LCL height around the monsoon onset is clear. But, in other sense, the Bowen ratio is a resulting parameter instead of a controlling variable. The authors need to be careful in describing the analyses.*

We are not sure what the reviewer has in mind here. Perhaps the issue is that monsoon precipitation causes the Bowen ratio to change and in this way the Bowen ratio is both the effect (say, on longer time scale) and the cause (say, on daily time scale) of the differences in convection and precipitation. Such thinking indeed brings the soil moisture – precipitation feedback. We revised the manuscript following such an argument and added a more detailed discussion below.

*Rev: 2) Regarding the soil moisture feedback, there are numerous literature that describes the soil-moisture-precipitation feedback processes (e.g., Asharaf et al. 2012, Soil Moisture–Precipitation Feedback Processes in the Indian Summer Monsoon Season). It is recommended to cite these papers in explaining the physical mechanism, and an addition of a new insight from the previous literature.*

We agree that a feedback mechanism exists between LCL heights and surface energy balance partitioning. However, land surface parameters such as soil moisture, vegetation cover etc., collectively determine energy balance partitioning, which then influences turbulent motions and boundary layer depth (Jones et al, 2009). Arguably, of all the surface properties, soil moisture has the largest impact on Bowen ratio. Soil moisture has the memory of atmospheric processes (Orlowsky et

al, 2009); it responds to precipitation variability and affects precipitation through evaporation (Douville et al, 2010). This soil moisture – precipitation (S-P) feedback has been extensively studied in the past. Asharaf et al, 2011 studied the S-P feedback for the Indian summer monsoon and found that premonsoon soil moisture has a significant influence on the monsoonal precipitation.

Arguably, the memory effect dominates on a scale of several days. But for a single day, Bowen ratio can act as the controlling factor rather than the consequence. This has been demonstrated in several studies. Rabin et al (1990) studied the observed variability of clouds over a landscape using a one dimensional parcel model, attributing it to the changes in Bowen ratio. This study points to a previous finding by Rabin (1977) which states that on moist days clouds develop earlier over places with low Bowen ratio, and on dry days convection occurs sooner over regions with higher Bowen ratio. Lewellen et al (1996) studied the role of Bowen ratio in determining the structure of boundary layer clouds using Large Eddy Simulations (LES). The study suggests lower cloud ceilings for low values of Bowen ratio. Schar et al, 1998 conducted simulations using a regional climate model on S-P feedback and stated that wet soils with small Bowen ratio produces shallow BL where the fluxes of heat and moisture are concentrated in a small volume of air. This leads to the build-up of high levels of moist entropy and provides a source for convective instability. Simulations also suggested that level of free convection was lower over wet soils.

As stated above, we included elements of this discussion and selected references into the revised manuscript.

References mentioned above:

Asharaf Shakeel, Andreas Dobler and Bodo Ahrens, 'Soil Moisture–Precipitation Feedback Processes in the Indian Summer Monsoon Season', 2012, Journal of Hydrometeorology, DOI: 10.1175/JHM-D-12-06.1

Orlowsky Boris and Sonia I Senevirante, 'Statistical Analyses of Land–Atmosphere Feedbacks and Their Possible Pitfalls', 2010,Journal of Climate, DOI: 10.1175/2010JCLI3366.1

Douville H, Chauvin F and Broqua H, 'Influence of Soil Moisture on the Asian and African Monsoons. Part I: Mean Monsoon and Daily Precipitation', 2001, Journal of Climate, https://doi.org/10.1175/1520-0442(2001)014<2381:IOSMOT>2.0.CO;2

 Jones Aubrey R and Nathaniel A. Brunsell, 'Energy Balance Partitioning and Net Radiation Controls on Soil Moisture–Precipitation Feedbacks', 2009, Earth Interactions, Volume 13, /doi/pdf/10.1175/2009EI270.1

Lewellen D.C and Lewellen  W. S, 'Influence of Bowen Ratio on Boundary-Layer Cloud Structure', J.Atmos. Sci, 1996. https://doi.org/10.1175/1520-0442(2001)014<2381:IOSMOT>2.0.CO;2

Rabin RM, Stensrud DJ, Stadler S, Wetzel PJ, Gregory M.  1990. ' Observed effects of landscape variability on convective clouds' Bull. Am. Meteorol. Soc. 71(3): 272–280. https://doi.org/10.1175/1520-0477(1990)071<0272:OEOLVO>2.0.CO;2.

Rabin, R. M. 1977. The surface energy budget of a summer convective period. Master of Science Thesis, McGill University, Montreal, Canada, 125 pp.

Schar Christoph, Daniel Luthi, Urs Beyerle and Erdmann Heise 'The Soil–Precipitation Feedback: A Process Study with a Regional Climate Model',1999, Journal of Climate, *https://doi.org/10.1175/1520-0442(1999)012<0722:TSPFAP>2.0.CO;2*